# Clinical profile, prognosis and post COVID-19 syndrome among UNRWA staff in Jordan: A clinical case-series study

**Haneen Aldahleh**[1]*, **Anwar Batieha**[2†], **Rasheed Elayyan**[2], **Nour Abdo**[2], **Ishtaiwi Abuzayed**[1], **Shatha Albaik**[3], **Yousef Shahin**[3], **Akihiro Seita**[3]

**1** Health department UNRWA, Jordan field office, Amman, Jordan, **2** Jordan University of Science and Technology, Irbid, Jordan, **3** Health department, UNRWA, Headquarters, Amman, Jordan

† Deceased.
* Haneen_yd@yahoo.com

**Data Availability Statement:** Data access requests can be sent to Mr. Sean McGreevy the Research Review Board (RRB) secretary at UNRWA by using the details below Sean McGreevy tel: +962 6 580

## Abstract

### Background

The clinical manifestations of Corona Virus Disease of 2019 (COVID-19) varied from patient to patient with evidence of multi-organ involvement. Many patients continue to have a wide range of symptoms for variable periods of time. The long-term effects of COVID-19 infection (post COVID-19 illness or syndrome) are not yet fully explored. This study aims to shed light on the clinical manifestations of the acute COVID-19 infection as well as post COVID-19 syndrome among the United Nations Relief and Works Agency for Palestine Refugee (UNRWA) staff in Jordan.

### Methods

A clinical case-series was conducted on a sample of COVID-19 positive employees of the UNRWA staff in Jordan. A structured questionnaire based mainly on World Health Organization (WHO) Case Report Form (CRF) verified tool for post COVID-19 was used. A sample of 366 out of a total of 1322 confirmed cases was systemically selected and included in the present study. Data were collected from UNRWA medical records and phone interviews. Data were analyzed using the Statistical Package for Social Sciences (SPSS) software.

### Results

The calculated Case Fatality Ratio was 0.7%. The incidence of COVID-19 among UNRWA staff in Jordan during the period of our study was 20.1%. A total of 366 respondents, 220 (60.1%) females and 146 (39.9%) males were included in the study. The mean (SD) age was 44.2 (8.0) years. Most of the infected (97.8%) developed acute COVID-19 symptoms. Fatigue, fever, joint pain, loss of smell and taste, and cough were the most common symptoms. According to WHO clinical classification of acute illness severity, 65.0% had mild illness. Only 28.7% of all subjects fully recovered from the infection, while most of them (71.3%) continued to suffer from many symptoms. Persistent fatigue (39.7%), shortness of

8495 | mobile: +962 790 900118 e-mail:
s.mcgreevy@unrwa.org.

**Funding:** The authors received no specific funding
for this work.

**Competing interests:** The authors have declared
that no competing interests exist.

**Abbreviations:** BMI, Body Mass Inde; COVID-19,
Corona Virus Disease of 2019; CRF, Case Report
Form; FLoD, First Line of Defense; ICU, Intensive
Care Unit; IRB, Institutional Review Board; JUST,
Jordan University of Science and Technology;
RRB, Research Review Board; SARS-CoV-2,
Severe Acute Respiratory Syndrome Coronavirus-
2; SOB, Shortness of Breath; SPSS, Statistical
Package for Social Sciences; UNRWA, United
Nations Relief and Works Agency for Palestine
Refugee; WHO, World Health Organization.

breath (SOB) with activity (18.8%), anxiety (17.4%), forgetfulness (16.9%), trouble in concentrating (16.7%), and depressed mood (15.8%) were the most frequently reported.

## Conclusion

Post COVID-19 illness was very common (71.3%) calling for UNRWA to continue assessment of post COVID-19 syndrome and the medical and psychological needs of affected staff. Despite vaccination, only 2.2% of the infected were asymptomatic. Reinfection was unusually high (24%).

## Background

In late 2019, an unexplained pneumonia caused by a new pathogenic virus called Severe Acute Respiratory Syndrome Coronavirus-2 (SARS-CoV-2) was reported in Wuhan, China. COVID-19 was the name given to this contagious disease [1]. WHO considered COVID-19 as a global pandemic on 11 March 2020 [2].

The clinical course and consequences of the disease are still not fully understood. The COVID-19 has a wide clinical spectrum from being asymptomatic to fatal [3].

Some patients recover completely, but many continue to have a variety of persistent symptoms long after the acute infection, and this is termed Post COVID-19 Syndrome or Long COVID-19 condition [4].

The proportion of persons who develop post COVID-19 syndrome differs from one study to another. Early published reports showed that about 10 to 20 percent of COVID-19 patients have symptoms that last for weeks to months after the infection [5]. A study mentioned that one out of every seven COVID-19 patients was still symptomatic after 12 weeks based on the available evidence [6]. Later reports showed a much higher percentage, A study from Saudi Arabia at the national level, showed that post COVID-19 symptoms were reported by 79.4% of individuals [7].

In Jordan, A recently published study by Al-Husinat, et al, used The Newcastle Post COVID-19 Syndrome Follow p Screening Questionnaire on 495 participants, showed that 83% of patients in the entire group had at least one post COVID-19 symptom, and 33.9% of participants had at least one symptom that persisted after mild or moderate SARS-CoV-2 infection, [8]. Another study on 657 patients who had recovered from SARS-CoV-2 infection at least after 3 months of the disease onset, post COVID-19 syndrome prevalence was reported to be 71.9% [9]. These figures highlight the importance of identifying the burden of post COVID-19 syndrome and the needs of affected individuals in order to effectively plan healthcare services and allocate public health resources [10].

In our study we focus on the clinical profile of patients during and after the episode among UNRWA staff. At the agency level, identifying the incidence of post COVID-19 syndrome and the medical and psychological needs of affected staff is very important to avoid their suffering and to ensure that work continues smoothly without interruption.

## Methods

### Study design, participants and settings

A clinical case-series study was conducted during the period February 15, 2022, to March 24, 2022. The study population included COVID-19 positive employees of the UNRWA in Jordan who were diagnosed during the period September 23, 2020, to November 23, 2021. The whole

UNRWA staff in Jordan is approximately 6572 employees, distributed in different departments (health, education, relief and social services, and microfinance).

## Sampling and data collection

The UNRWA First Line of Defense (FLoD) team for COVID-19 has a database including the confirmed COVID-19 cases among the staff, a total of 1322 employees, as of November 23, 2021. All these patients confirmed to be positive by Polymerase Chain Reaction test for COVID19. All infected employees with less than 12 weeks since their diagnosis—as the post COVID-19 syndrome according to WHO clinical case definition is the persistence of symptoms beyond 12 weeks from the date of onset-, and those who died (9 employees) were excluded from the selection process. We didn't have the contact details of some of these employees at time of our study, as they were out of duty, so they were also excluded. We selected our sample systematically, with a sample interval of 3. Our sample size was 366.

The data collection process was performed based on:

**1. UNRWA Medical Records/Data Base.**   The UNRWA medical records were abstracted on a special form prepared for the purpose of this study. Relevant data included: Contacted details, Age, Name, Sex, Date of diagnosis, Date of recovery, Vaccination status, and others

**2. Phone Interviews.**   Phone Interviews were conducted by the researcher with all members of the sample, a total of 366 patients. The response rate was 100%

## Study tool

A structured questionnaire prepared for the purpose of this study and based mainly on WHO Case Report Form (CRF) verified tool for post COVID-19 was used. It was available in English and Arabic. Some of the questions were deleted from the questionnaire after conducting a pilot study on a sample of 30 participants, these questions were mostly answered as unknown. Also, a minimal change made on the format and wording of the translated questions based on participants' feedback. Otherwise, the questionnaire found to be clear, comprehensive, provide complete and accurate information, and simple to be used by interviewers and respondents.

Our questionnaire consisted of four sections, distributed as follows:

**1. Sociodemographic Status.**   This section consisted of questions related to sociodemographic factors which included (age, sex, pregnancy information, highest education level, smoking status, BMI, job details), in addition to history of admission to a health care facility or a hospital in the past 3 years.

**2. Clinical Picture and Disease Progression.**   This part included chronic comorbidities, medications history, responders' compliance with preventive measures, the diagnostic test used to confirm the diagnosis, and the source of infection. Then a section covered the details of the acute infection including the symptoms, the severity of acute illness based mainly on the need of respiratory support, and the level of its need, complications, highest level of care, and the treatment during the acute illness. In addition to that a history of reinfection was recorded.

**3. Vaccination Status.**   This section encompassed vaccination status, the number of doses, the specific time of each dose, and the type of vaccine.

**4. Residual Symptoms after Recovery Study.**   This section was created to assess patients' status after recovery, outlining the possibility of Post COVID-19 illness development. Patients were asked about 50 possible symptoms according to WHO CRF form. Key symptoms were categorized according to their presence at time of questioning as: was present and now resolved, still present, not present, sometimes present and unknown. Symptoms were considered positive only if persisted for two months or more after 12 weeks from date of infection

and cannot be explained by an alternative diagnosis. Functioning, ability of self-care, and occupational changes were also assessed.

## Data management and statistical analysis

Data entry was performed using Microsoft Excel 2016 sheets, and data analysis was performed using the Statistical Package for Social Sciences (SPSS) software version 25. We conducted logical and range checks to check for errors in data entry; detected errors were addressed by returning to the original questionnaire; errors that were hard to correct were considered as missing data.

We obtained descriptive statistics for relevant variables as appropriate. Continuous variables were described by their means and SDs, while categorical variables were presented with their frequencies and percentages for each category.

The Chi-square test was used to test the significance of relationships between categorical variables. A p-value of less than 0.05 was considered statistically significant.

## Data security

The data collected used a platform that was prepared specifically for this purpose. This platform was protected using a strong password, it was only accessed using specific UNRWA protected devices, and if it was necessary to use portable devices for initial collection or storage of identifiers, the data files were encrypted, and the identifiers moved to a secure system as soon as possible after collection. The identifiable data is limited to members of the study team including the researchers, the study supervisor, and UNRWA COVID-19 technical advisor.

**Ethical considerations.** The study was approved by the Institutional Review Board (IRB) of the Jordan University of Science and Technology (JUST) and Research Review Board (RRB) of UNRWA. An informed verbal consent was obtained from all patients. Identifying information were kept strictly confidential and the data was used only by the investigator for scientific purposes. Participation was voluntary and the participants had their full freedom to leave the study at any stage. The study carried no foreseeable harm to participants as it relies on conducting interviews with no invasive techniques.

**Definition of post COVID-19 syndrome.** The WHO clinical case definition of post COVID-19 syndrome was used in the present study, which refers to the condition which occurs in individuals with a probable or confirmed SARS-COV-2, three months after the onset of COVID-19, with symptoms that continue at least for two months and cannot be explained by an alternative diagnosis.

## Results

### Characteristics of the study population

The incidence of COVID-19 among UNRWA staff in Jordan during the period from September 23,2020 to November 23,2021 was 20.1% (1322 confirmed cases out of 6572 employees), and the calculated Case Fatality Ratio was 0.7% (9 deaths out of 1322 confirmed cases).

As shown in Table 1, the percentage of females in the study was higher than that of males (220 of our patients were females (60.1%) and 146 (39.9%) were males). The vast majority has a college or higher degree education (92.7%). About 15.3% of our patients were health care workers. Age ranged from 26 to 60 years, with a mean (SD) of 44.2 (8.0) years. The mean (SD) BMI was 28.4 (4.5) kg/m2; approximately 32.8% were obese, and 45.1% overweight. Around 19.1% of our patients were current smokers and 3.6% were former smokers. About 40.2% of our patients were healthy (free of comorbidities). Obesity (32.8%) and hypertension (20.2%) were the most frequently encountered comorbidities among infected UNRWA employees in Jordan.

**Table 1. Sociodemographic characteristics and comorbidities among infected UNRWA employees in Jordan, 2021.**

| Category | Variable | N (%) |
|---|---|---|
| **Sex** | Male | 146 (39.9) |
| | Female | 220 (60.1) |
| **Age groups (years)** | 20–39 | 119 (32.5) |
| | 40–60 | 247 (67.5) |
| **BMI (Kg/m2)** | **Normal weight**: (BMI 18.5–24.9) | 81 (22.13) |
| | **Overweight:** (BMI 25–29.9) | 165 (45.10) |
| | **Obesity class I**: (BMI 30–34.9) | 92 (25.14) |
| | **Obesity class II**: (BMI 35–39.9) | 22 (6.00) |
| | **Obesity class III**: (BMI > 40) | 6 (1.63) |
| | *WHO Classification | |
| **Highest level of education** | Middle school or lower | 27 (7.3) |
| | College or higher | 339 (92.7) |
| **Smoking status** | Current smoker | 70 (19.1) |
| | Former smoker | 13 (3.6) |
| | Never smoker | 283 (77.3) |
| **Department** | Education | 266 (72.7) |
| | Health | 56 (15.3) |
| | Others | 44 (12.0) |
| **Comorbidities Status** | Free of comorbidities | 147 (40.2) |
| | One Comorbidity | 141 (38.5) |
| | Two or more comorbidities | 78 (21.3) |
| **Comorbidity** | Obesity > 30 | 120 (32.8) |
| | Hypertension | 74 (20.2) |
| | Diabetes | 29 (7.9) |
| | Hyperlipidemia | 25 (6.8) |
| | Hypothyroidism | 20 (5.4) |
| | Chronic heart disease (not hypertension) | 12 (3.3) |
| | Immunodeficiency | 9 (2.5) |
| | Chronic lung disease | 6 (1.6) |
| | Chronic neurological disorders | 4 (1.1) |
| | Cancer | 2 (0.5) |
| | Asplenia | 1 (0.3) |
| | Chronic kidney disease | 1 (0.3) |
| | Mental health condition | 1 (0.3) |
| | Others | 42 (11.5) |

## Adherence to preventive measures and the source of COVID-19 infection

As shown in Tables 2 and 3, most of our patients adhered to COVID-19 preventive measures. However; 38% of the patients believed they contracted the disease from contact with patients at job and 30.3% from infected family members.

## Symptoms during the acute phase of the disease

As shown in Fig 1, within the acute phase of the disease, the most frequently encountered acute symptoms were fatigue (76.2%), smell impairment (73%), fever (71.6%), taste

**Table 2. Adherence to preventive measures among infected UNRWA employees in Jordan, 2021.**

| Preventive measure | Yes N (%) | Sometimes N (%) | No N (%) |
|---|---|---|---|
| **Adherence to mask** | 303 (82.8) | 63 (17.2) | 0 (0.0) |
| **Adherence to physical distancing** | 309 (84.4) | 56 (15.3) | 1 (0.3) |
| **Washing hands frequently** | 322 (88.0) | 44 (12.0) | 0 (0.0) |

**Table 3. Source of COVID-19 infection among infected UNRWA employees in Jordan, 2021.**

| Source of infection | N | % |
|---|---|---|
| A case or contact at job | 139 | 38.0 |
| A case or contact at home | 111 | 30.3 |
| Unknown | 90 | 24.6 |
| Crowded places | 17 | 4.6 |
| Health facilities | 8 | 2.2 |
| From outside the country | 1 | 0.3 |
| Total | 366 | 100.0 |

impairment (68.6%), joint/bone pain (63.7%), and cough (56.0%). Only 8 participants (2.2%) were completely asymptomatic.

## Severity of acute disease and highest level of care needed

The severity of acute illness of COVID-19 was graded according to WHO criteria for severity, which was illustrated in the CRF verified tool for post COVID-19, and based mainly on the need of respiratory support, and the level of its need.

As shown in Table 4 and Fig 2, most of our patients (65%) experienced a mild form of acute illness, and only 15 patients were admitted to hospital (4.1%), of whom 4 (26.7%) were admitted to ICU (Intensive Care Unit), 3 of them (75%) needed invasive ventilation.

## Vaccination status

At the time of our study about 98.9% of our patients had received at least one dose of a COVID-19 vaccine, 88% received 2 doses, and 10.4% received 3 doses (Table 5).

As shown in Table 6, only 22.1% of our patients received at least 2 doses of vaccine prior to their infection, and among those only 4 patients (5%) developed severe form of infection, and none of them developed a critical disease.

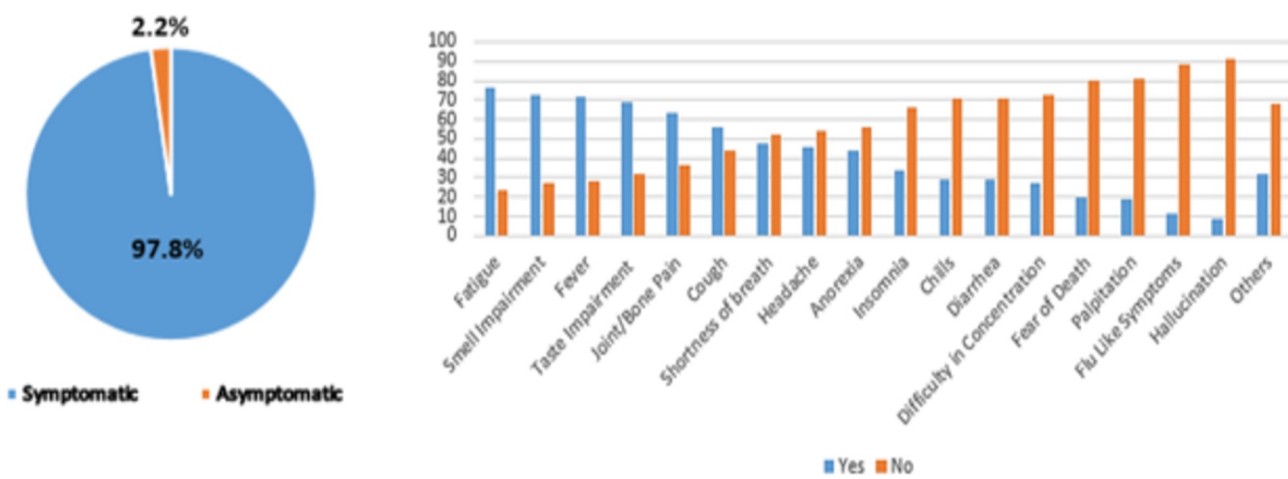

**Fig 1. Symptoms presence and the most common symptoms encountered during the acute phase of COVID-19.**

**Table 4. Highest level of care, severity, and need of oxygen during the acute phase of COVID-19.**

| Variable | Outcome | N | % |
|---|---|---|---|
| **Highest level of care** | Self-care | 171 | 46.7 |
| | Outpatient | 155 | 42.4 |
| | Telemedicine | 25 | 6.8 |
| | Hospital admission | 15 | 4.1 |
| | **Total** | **366** | **100.0** |
| **Severity of acute illness** | Mild | 238 | 65.0 |
| | Moderate | 105 | 28.7 |
| | Severe | 20 | 5.5 |
| | Critical | 3 | 0.8 |
| | **Total** | **366** | **100.0** |
| **Oxygen therapy** | No | 343 | 93.7 |
| | Non-invasive | 20 | 5.5 |
| | Invasive ventilation | 3 | 0.8 |
| | **Total** | **366** | **100.0** |

Patients who received < 2 doses of vaccination tended to have milder form of the disease (67%) compared to patients who received ≥ 2 doses of the vaccine prior to infection (58%), but the association was not statistically significant, (P = 0.239) (Table 6).

## Reinfection

About one quarter (N = 88, 24%) of the study sample experienced a second episode of infection, two of whom experienced a third one.

## Post COVID-19 illness

**Presence of post COVID-19 manifestations.** All our patients were selected at least 12 weeks after their infection and asked about residual or newly developed symptoms. As shown in Table 7, Only 28.7% of them were completely free of symptoms, while 71.3% experienced at

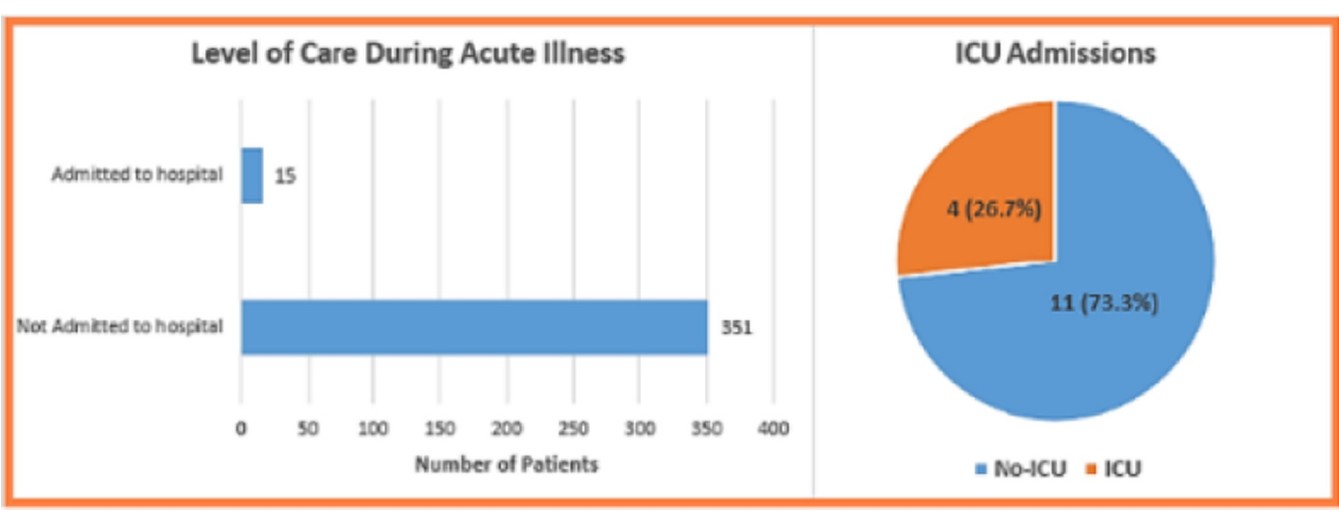

**Fig 2. Level of care during acute illness, and ICU admission status.**

**Table 5. Vaccination status at the time of our study among infected UNRWA employees in Jordan, 2021.**

| Vaccination status at the time of our study | N | % |
|---|---|---|
| Not vaccinated | 4 | 1.1 |
| Received 1 dose | 2 | 0.5 |
| Received 2 doses | 322 | 88.0 |
| Received 3 doses | 38 | 10.4 |
| Total | 366 | 100.0 |

least one or more of the Post COVID-19 symptoms studied, with a follow up median (mean) of 362 (346) days from the initial diagnosis.

**Post COVID-19 symptoms.** A detailed review of more than 50 possible residual symptoms was conducted during our patients' interviews. As shown in Table 8, among those symptoms, Persistent fatigue (39.7%) was the most common symptom encountered, followed by SOB with activity (18.8%), Anxiety (17.4%), Forgetfulness (16.9%), Trouble in concentrating (16.7%), and Depressed mood (15.8%).

**Presence of post COVID-19 symptoms in relation to the time of acute illness.** As shown in Table 9, about three quarters (75.8%) of those who had post COVID-19 symptoms, had been diagnosed for more than 6 months.

## Association of the most common post COVID-19 symptoms and relevant study variables

We examined the data for associations between the six most common post COVID-19 symptoms (Persistent fatigue, SOB with activity, anxiety, forgetfulness, trouble in concentration, and depressed mood) and selected study variables, to identify the possibility of high-risk subgroups.

**Post COVID-19 symptoms by gender.** As shown in Table 10, anxiety was significantly higher in females (21.4%) than in males (11.6%) (P = 0.017). Persistent fatigue was higher in females (42.7%) than in males (34.9%) with no significant association (P = 0.135). Also, forgetfulness, trouble in concentration and depressed mood were higher in females than males with no significant association (P = 0.437, 0.504, 0.133, respectively)

**Table 6. Association between the severity of acute illness and having received two doses of the vaccine prior to infection.**

| Received at least 2 doses of vaccine prior to acute infection | N (%) | Severity of acute illness | | | | P-Value |
|---|---|---|---|---|---|---|
| | | Mild N (%) | Moderate N (%) | Severe N (%) | Critical N (%) | |
| No | 285 (77.9) | 191 (67.0) | 75 (26.3) | 16 (5.6) | 3 (1.1) | 0.239 |
| Yes | 81 (22.1) | 47 (58.0) | 30 (37.0) | 4 (5.0) | 0 (0.0) | |
| Total | 366 (100) | 238 (65.0) | 105 (28.7) | 20 (5.5) | 3 (0.8) | |

**Table 7. Presence of post COVID-19 symptoms among infected UNRWA employees in Jordan, 2021.**

| Do you still have symptoms? | N | % |
|---|---|---|
| No | 105 | 28.7 |
| Yes | 261 | 71.3 |
| Total | 366 | 100.0 |

**Table 8. Post COVID-19 symptoms among infected UNRWA employees in Jordan, 2021.**

| Do you still have symptoms? | No | | Yes, (Had, now resolved) | | Yes, (Currently) | |
|---|---|---|---|---|---|---|
| | N | % | N | % | N | % |
| Persistent fatigue | 203 | 55.5 | 18 | 4.9 | 145 | 39.7 |
| Shortness of breath with activity | 285 | 77.9 | 12 | 3.3 | 69 | 18.8 |
| Anxiety | 289 | 79.0 | 13 | 3.6 | 64 | 17.4 |
| Forgetfulness | 303 | 82.8 | 1 | 0.3 | 62 | 16.9 |
| Trouble in concentrating | 299 | 81.7 | 6 | 1.6 | 61 | 16.7 |
| Depressed mood | 291 | 79.5 | 17 | 4.6 | 58 | 15.8 |
| Joint pain/swelling | 307 | 83.9 | 6 | 1.6 | 53 | 14.5 |
| Altered smell | 294 | 80.3 | 37 | 10.1 | 35 | 9.5 |
| Altered taste | 303 | 82.8 | 30 | 8.2 | 33 | 9.0 |
| Problems with communication | 332 | 90.7 | 4 | 1.1 | 30 | 8.2 |
| Dysmenorrhea | 333 | 91.0 | 7 | 1.9 | 26 | 7.1 |
| Sleeping less | 338 | 92.3 | 2 | 0.5 | 26 | 7.1 |
| Sleeping more | 341 | 93.2 | 2 | 0.5 | 23 | 6.3 |
| Loss of interest/pleasure | 342 | 93.4 | 1 | 0.3 | 23 | 6.3 |
| Shortness of breath at rest | 344 | 94.0 | 2 | 0.5 | 20 | 5.4 |
| Persistent dry cough | 332 | 90.7 | 15 | 4.1 | 19 | 5.2 |
| Palpitation | 345 | 94.3 | 2 | 0.5 | 19 | 5.2 |
| Chest pain | 345 | 94.3 | 3 | 0.8 | 18 | 4.9 |
| Post-exertional malaise | 347 | 94.8 | 3 | 0.8 | 16 | 4.4 |
| Persistent muscle pain | 347 | 94.8 | 4 | 1.1 | 15 | 4.1 |
| Stomach pain | 352 | 96.2 | 1 | 0.3 | 13 | 3.6 |
| Persistent headache | 350 | 95.6 | 3 | 0.8 | 13 | 3.5 |
| Erectile dysfunction | 354 | 96.7 | 1 | 0.3 | 11 | 3.0 |
| Ringing in ears | 355 | 97.0 | 2 | 0.5 | 9 | 2.5 |
| Weight gain | 355 | 97.3 | 1 | 0.3 | 9 | 2.5 |
| Dizziness/ lightheadedness | 356 | 97.3 | 1 | 0.3 | 9 | 2.5 |
| Behavior change | 357 | 97.5 | 0 | 0 | 9 | 2.4 |
| Diarrhea | 358 | 97.8 | 1 | 0.3 | 7 | 1.9 |
| Weakness in limbs | 359 | 98.1 | 1 | 0.3 | 6 | 1.7 |
| Loss of appetite | 359 | 98.1 | 1 | 0.3 | 6 | 1.6 |
| Problems seeing | 361 | 98.6 | 0 | 0 | 5 | 1.4 |
| Problems hearing | 360 | 98.4 | 1 | 0.3 | 5 | 1.4 |
| Problems with balance | 361 | 98.6 | 0 | 0 | 5 | 1.4 |
| Weight loss | 360 | 98.4 | 2 | 0.5 | 4 | 1.1 |
| Constipation | 362 | 98.9 | 0 | 0 | 4 | 1.1 |
| Stiffness of muscles | 360 | 98.4 | 2 | 0.5 | 4 | 1.1 |
| Swollen ankles | 362 | 98.9 | 0 | 0 | 4 | 1.0 |
| Problem swallowing | 363 | 99.2 | 0 | 0 | 3 | 0.8 |
| Skin rash | 361 | 98.6 | 2 | 0.5 | 3 | 0.8 |
| Problems passing urine | 363 | 99.2 | 1 | 0.3 | 2 | 0.5 |
| Lumpy lesions: (purple/pink/bluish) on toes/COVID toes | 364 | 99.5 | 1 | 0.3 | 1 | 0.3 |
| Jerking of limbs | 365 | 99.7 | 0 | 0 | 1 | 0.3 |
| Fever | 362 | 98.9 | 3 | 0.8 | 1 | 0.3 |
| Tremors | 364 | 99.5 | 1 | 0.3 | 1 | 0.3 |
| Can't move and/or feel one side of body or face | 365 | 99.7 | 0 | 0 | 1 | 0.3 |
| Problems with gait/falls | 365 | 99.7 | 0 | 0 | 1 | 0.3 |
| Slowness of movement | 365 | 99.7 | 0 | 0 | 1 | 0.3 |

**Table 9. Post COVID-19 symptoms time frame among infected UNRWA employees in Jordan, 2021.**

| Post COVID-19 Symptoms | ≥ 3 to ≤ 6 months N (%) | > 6 to ≤ 12 months N (%) | > 12 months N (%) | Total N (%) |
|---|---|---|---|---|
| **No** | 24 (22.9) | 40 (38.1) | 41 (39) | 105 (28.7) |
| **Yes** | 63 (24.1) | 104 (39.8) | 94 (36.0) | 261 (71.3) |

**Post COVID-19 symptoms by smoking status.** A significant association was found between forgetfulness as a post COVID-19 symptom and smoking (P = 0.015). About 28.6% of smokers, 15.4% of former, and 14.1% of those who were never smokers found to have forgetfulness. A high percent of smokers (44.3%) had persistent fatigue as a post COVID-19 symptom (P = 0.674). No significant association between smoking status and SOB with activity, anxiety, trouble in concentration and depressed mood (P = 0.924, P = 0.555, P = 0.317, P = 0.242) (Table 10).

**Post COVID-19 symptoms by comorbidities.** As shown in Table 10, persistent fatigue and SOB with activity were found to be significantly associated with obesity (P = 0.009, 0.017, respectively). Also, anxiety, forgetfulness, trouble in concentration and depressed mood were higher in obese patients than non-obese patients with no significant association (P = 0.377, 0.842, 0.765, 0.363, respectively).

Depressed mood was found to be higher in diabetic patients (31.0%) than non-diabetic patients (14.5%) with a significant association (P = 0.020). Also, persistent fatigue, SOB with activity, anxiety and forgetfulness were higher in diabetic patients than non-diabetic patients with no significant association (P = 0.320, 0.792, 0.136, 0.111, respectively). Trouble in concentration was found to be higher in non-diabetic patients (17.8%) than in diabetic patients (3.4%) with a significant association (P = 0.047).

SOB with activity, anxiety, and forgetfulness were found to be higher in hypertensive patients than in non-hypertensive patients with no significant association (P = 0.093, 0.984, 0.611, respectively). Persistent fatigue, trouble in concentration, and depressed mood were found to be higher in non-hypertensive patients than in hypertensive patients with no significant association (P = 0.377, 0.907, 0.796, respectively).

**Table 10. Association between the most common post COVID-19 symptoms and selected study variables.**

| Post COVID-19 symptom | Sex | | | Smoking status | | | | Comorbidities | | | | | | | | |
|---|---|---|---|---|---|---|---|---|---|---|---|---|---|---|---|---|
| | Male N (%) | Female N (%) | P-Value | Current smoker N (%) | Former N (%) | Never smoker N (%) | P-Value | Hypertension N (%) | | | Diabetes N (%) | | | Obesity N (%) | | |
| | | | | | | | | No | Yes | P-Value | No | Yes | P-Value | No | Yes | P-Value |
| **Persistent fatigue** | 51 (34.9) | 94 (42.7) | 0.135 | 31 (44.3) | 5 (38.5) | 109 (38.5) | 0.674 | 119 (40.8) | 26 (35.1) | 0.377 | 131 (38.9) | 14 (48.3) | 0.320 | 40 (16.3) | 24 (20.0) | 0.009 |
| **SOB with activity** | 29 (19.9) | 40 (18.2) | 0.687 | 13 (18.6) | 3 (23.1) | 53 (18.7) | 0.924 | 50 (17.1) | 19 (25.7) | 0.093 | 63 (18.7) | 6 (20.7) | 0.792 | 38 (15.4) | 31 (25.8) | 0.017 |
| **Anxiety** | 17 (11.6) | 47 (21.4) | 0.017 | 14 (20.0) | 1 (7.7) | 49 (17.3) | 0.555 | 51 (17.5) | 13 (17.6) | 0.984 | 56 (16.6) | 8 (27.6) | 0.136 | 40 (16.3) | 24 (20.0) | 0.377 |
| **Forgetfulness** | 22 (15.1) | 40 (18.2) | 0.437 | 20 (28.6) | 2 (15.4) | 40 (14.1) | 0.015 | 48 (16.4) | 14 (18.9) | 0.611 | 54 (16.0) | 8 (27.6) | 0.111 | 41 (16.7) | 21 (17.5) | 0.842 |
| **Trouble in concentration** | 22 (15.1) | 39 (17.7) | 0.504 | 13 (18.6) | 4 (30.8) | 44 (15.5) | 0.317 | 49 (16.8) | 12 (16.2) | 0.907 | 60 (17.8) | 1 (3.4) | 0.047 | 40 (16.3) | 21 (17.5) | 0.765 |
| **Depressed mood** | 18 (12.3) | 40 (18.2) | 0.133 | 13 (18.6) | 0 (0.0) | 45 (15.9) | 0.242 | 47 (16.1) | 11 (14.9) | 0.796 | 49 (14.5) | 9 (31.0) | 0.020 | 36 (14.6) | 22 (18.3) | 0.363 |

SOB with activity was higher in males (19.9%) than in females (18.2%) with no significant association (P = 0.687)

**Table 11. The association between the most common post COVID-19 symptoms and losing a friend or a relative in the pandemic.**

| Post COVID-19 symptom | Losing a friend or a relative in the pandemic | | P-Value |
|---|---|---|---|
| | Yes | No | |
| **Persistent fatigue** | 71 (47.3) | 74 (34.3) | 0.012 |
| **SOB with activity** | 36 (24.0) | 33 (15.3) | 0.036 |
| **Anxiety** | 34 (22.7) | 30 (13.9) | 0.030 |
| **Forgetfulness** | 22 (14.7) | 40 (18.5) | 0.334 |
| **Trouble in concentration** | 18 (12.0) | 43 (19.9) | 0.046 |
| **Depressed mood** | 31 (20.7) | 27 (12.5) | 0.035 |

**Post COVID-19 symptoms by losing a friend or a relative in the pandemic.** The results indicated that there is a significant association between persistent fatigue, SOB with activity, anxiety, and depressed mood and losing a friend or a relative in the pandemic (P = 0.012, 0.036, 0.030, 0,035). Forgetfulness and trouble in concentration were found to be higher in those who didn't lose a friend or relative in the pandemic (P = 0.334, 0.046, respectively) (Table 11).

## Discussion

The present study was conducted on infected UNRWA staff in Jordan to evaluate their journey with COVID-19, during and after the acute phase of infection.

In this section we discuss the main findings of the study in relation to available literature. Also, we outlined the limitations and strengths of the study. It should be kept in mind that our results pertain to a working group (UNRWA employees, all ≤60 years of age) and, thus, older people over the age of 60 are not represented in our study. The vast majority are well-educated with about 93% having a college education or higher.

### Acute phase of COVID-19

The incidence of COVID-19 among UNRWA staff in Jordan during the period from September 23,2020 to November 23,2021 was 20.1%, and the calculated Case Fatality Ratio was 0.7%.

Our present study revealed a very low percentage (2.2%) of asymptomatic patients during the acute infection phase. Compared to literature, a meta-analysis of 41 studies conducted on 50155 COVID-19 patients, found that the pooled prevalence of having asymptomatic infection was 15.6% [11]. Another huge meta-analysis of 390 studies, involved 104,058 patients, showed the prevalence of truly asymptomatic COVID-19 patients to be 35.1% [12].

In a meta-analysis and a systematic review of 48 publications which included 24,410 patients with COVID-19 from 9 countries, the most frequent symptoms were fever (78%), cough (57%) and fatigue (31%) [13]. Studying 56 publications, 89% of them from China, the most common symptoms reported by the patients were fever, cough, shortness of breath, and fatigue [14]. A study from Jordan on 371 patients, in which 59% of them were symptomatic, the most common symptoms were generalized fatigue (51%), dry cough (45.8%), and fever (41.8%) [15]. In our study, the most frequently encountered acute symptoms were fatigue (76.2%), smell impairment (73%), fever (71.6%), taste impairment (68.6%), joint/bone pain (63.7%) and cough (56.0%).

It is evident that symptoms during the acute infection differed from one study to another with no clear explanation offered yet.

The symptomatic profile we observed, dominated by fatigue, smell impairment, and fever, resonates with international findings but with certain nuances. Such variations could stem

from differences in genetic predispositions, viral strains, or environmental factors, warranting further investigation.

## Vaccination and reinfection

Our study sample showed that 98.9% of employees received at least one dose of the vaccine. Such a high percentage may be explained by the regulations outlined by the government defense order which indicated that employees were not allowed to join their work without being fully vaccinated. Due to that it was not possible to study the infection rate among vaccinated and unvaccinated. However, we were able to show the association between the severity of acute illness and receiving 2 doses of the vaccine before getting the infection. Only 22.1% of our patients received at least 2 doses of vaccine prior to their infection, and among those only 4 patients (5%) developed severe form of infection, and none of them developed a critical disease.

Cases of SARS-Cov-2 reinfection were reported with increasing prevalence. About one quarter of our patients experienced reinfection confirmed by PCR testing. This may be overrated because we cannot explain if it was a persistent virus shedding from the previous infection, and consequently a persistent positive PCR testing, or it was a new infection. Moreover, the absence of virus sequencing prevented us from differentiating if it was the same virus strain or not.

## Post COVID-19 symptoms

COVID-19 recovery should not only depend on testing negative for SARS-COV-2 infection, or positivity for antibodies. Since the beginning of the pandemic, many people have experienced chronic symptoms and consequences following the acute phase of the disease. In literature, multiple terms are used to describe this condition, post-COVID-19 syndrome, long-term effects of COVID-19, post-acute COVID-19 syndrome, chronic COVID-19, long COVID-19 [16]. All of them indicate that the patient is still experiencing symptoms after acute COVID-19 infection. For convenience and clarity, we used the term Post COVID-19 syndrome in our study.

Our study showed that post COVID-19 syndrome is common, as 71.3% of the study sample experienced at least one of the post COVID-19 symptoms studied, with a follow up median (mean) of 362 (346) days from the initial diagnosis. In a study from Italy, a persistence of at least one post COVID-19 symptom after one year of follow up in non-hospitalized patients was 53% [17]. A systemic review and meta-analysis of 41 studies showed that the calculated global prevalence of the post-COVID-19 condition was 54% in hospitalized patients and 34% in non-hospitalized patients [18]. About 35.9% of 354 COVID-19 patients in France, had at least one Post COVID-19 symptom after a mean of 289.1 days from the initial diagnosis [19]. A study from Germany on a total of 1459 patients showed that the prevalence of post COVID-19 was 72.6% in hospitalized patients and 46.2% for non-hospitalized patients [20]. A recent study that was published in Jordan showed nearly the same results of our study, with 71.9% of patients experiencing at least one symptom of post COVID-19 [9]. As noticed, persistent COVID-19 symptoms have been experienced by most of patients, with a wide variation ranging from 35% and up to 90%. The inconsistency in the findings may be due different criteria, study design, and time frames used to define Post COVID-19 syndrome among the different studies. Regarding post COVID-19 symptoms, the present study showed more than 50 possible residual symptoms following acute infection by at least 12 weeks which persisted for at least two months. Among these symptoms, persistent fatigue (39.7%) was the mostly encountered, followed by shortness of breath with activity (18.8%), anxiety (17.4%), forgetfulness (16.9%), trouble in concentrating (16.7%), and depressed mood (15.8%). In a systematic review and meta-analysis of 15 studies, the 5 most common post COVID-19 symptoms

encountered were fatigue (58%,), headache (44%), attention disorder (27%), hair loss (25%), and dyspnea (24%) [21]. In the North of India, a study of 1234 patients who were followed up for a median of 91 days, the most common post COVID-19 symptoms were myalgia, fatigue, shortness of breath, cough, insomnia, mood disturbances, and anxiety [20]. Id *et al*. studied 128 patients who recovered from acute COVID-19, and found that 52.3% of them had a persistent fatigue as a common post COVID-19 syndrome at a median of 10 weeks after their initial diagnosis [22]. In a Jordanian study, the most common symptoms found to be dyspnea, fatigue, taste and smell impairment, cough, and depressed mood [9]. On a large meta-analysis of 29 peer reviewed publications and 4 preprints, the most common symptoms encountered were fatigue and shortness of breath, with a pooled prevalence ranging from 35 to 60% according to the follow up period [23]. Another large meta-analysis of 81 studies, reported a pooled prevalence of fatigue after 12 weeks of COVID-19 diagnosis of 30% [24].

As known, fatigue is one of the most prevalent symptoms experienced by patients during their acute SARS-COV-2 infection, as early reported in publications during the pandemic [22]. And based on the recent available literature, fatigue is the dominant residual feature after COVI19. For this reason, Collaborative on Fatigue Following Infection (COFFI) made recommendations for scientific clinical and research strategies, based on a systematic evaluation of fatigue following COVID-19 and other infectious diseases [25]. Because of its prolonged nature, others linked persistent fatigue after COVID-19 to what is called chronic fatigue syndrome or Myalgic Encephalomyelitis, which was described after serious infections such as SARS and MERS. As yet, there is no definitive pathophysiological clarification of post COVID-19 associated fatigue [26]. Anxiety, depressed mood, and trouble in concentration could be explained by patients' worries about their persistent symptoms and delay in recovery.

**Post COVID-19 by gender.**   In our study, females were more likely to develop most of post COVID-19 symptoms but without being statistically significant (P = 0.055). When the most common symptoms were studied separately, only anxiety showed a statically significant association with female sex (P = 0.017). In a meta-analysis and systematic review of 41 studies, post COVID-19 pooled prevalence was higher in females (49%) compared to that in males (37%) [18]. In Milan, Italy, a prospective cohort study on 377 patients found that females were 3 times more likely to have post COVID-19 syndrome than males [27]. Another multi-centric cohort study, also confirmed that female sex by itself was ≥3 more associated with post COVID-19 syndrome [28]. In Jordan a cross sectional study found that being female is a risk factor of post COVID-19 syndrome [9]. On the other hand, a study from Japan showed that sex had no significant association with post COVID-19 syndrome [29].

**Post COVID-19 by smoking status.**   Regarding post COVID-19 status and smoking, our study showed only a significant association between forgetfulness as a Post COVID-19 symptom and smoking (P = .015). In two meta-analyses, smoking was found to have a negative outcome in patients with COVID-19 [30, 31]. In Italy, Bai *et al*., in their study on 377 patients found that there was an association between post COVID-19 syndrome and smoking [27]. Another study from France, estimated that smoking is a risk factor for developing post COVID-19 symptoms, and mainly correlate with tachycardia and high blood pressure reported by patients [32].

**Post COVID-19 by comorbidities.**   Jordan National Stepwise Survey for non-communicable diseases in 2019 showed that the prevalence of hypertension is 22%. Among adults aged 45 to 69 year's old, diabetes was found in 20% of the population [33]. In our study, and because most of the respondents were in the working age group, the prevalence of comorbidities such as hypertension (20.2%) and diabetes mellitus (7.9%) were relatively low in comparison to those figures. However, more than three quarters of them were overweight or obese. In the present study, persistent fatigue and SOB with activity as post COVID-19 symptoms, were

found to be significantly associated with obesity, (P = 0.009, 0.017) respectively. It was found that those with obesity take a longer duration of time to get a clear chest radiograph, and this is consistent with its association with SOB with activity as a persistent symptom of post-acute infection [34]. A retrospective study on a total of 2839 patients, suggested that patients with obesity are at higher risk to develop post COVID-19 symptoms [35]. Ellen J. Thompson *et al.*, featured that post COVID-19 was found to be associated with pre-existing comorbidities and psychiatric conditions. Increased risk was encountered with a pre-pandemic presence of asthma and obesity but not with diabetes [36]. Fernandez-de-las-Penas *et al.*, found that diabetes was not a risk factor for post COVID-19 syndrome [37]. In our study, we found that depressed mood was significantly associated with being diabetic (P = 0.020) respectively.

While it's recognized that post-COVID symptoms might be influenced by factors like age, gender, or comorbidities, our study interestingly identified associations with smoking. This aligns with some international studies but diverges from others, hinting at potential genetic, behavioral, or regional factors at play.

## The strengths of the study

To our knowledge this is the first study to assess Post COVID-19 syndrome status in Jordan according to WHO case-report form. In our study, we attempted to select a representative sample of employees who got the infection at least 12 weeks before the assessment. As individuals working in an organized workplace (UNRWA), our patients were cooperative and easily followable. Thus, the response rate was 100%. All data were collected by the researcher who is a medical officer working at UNRWA, which adds to the credibility of the collected data. The study can serve as a prototype for other future studies at different working environments.

## Limitations

Findings of this study pertain to UNRWA employees whose age ranges from 20 to 60. Thus, the elderly and the young age groups are not represented in this study. The lack of sequencing information made it difficult to differentiate between recurrence and reinfection. Because almost all our patients were vaccinated, it was not possible to assess the vaccine effectiveness in prevention of the infection.

## Conclusion and recommendations

In conclusion, this study provides valuable insights into the prevalence and impact of post-COVID-19 syndrome among UNRWA staff in Jordan. The findings reveal that a significant proportion of staff who have recovered from COVID-19 are experiencing persistent symptoms that can affect their daily lives and work performance.

Such study can help to inform policies and interventions aimed at mitigating the long-term effects of COVID-19 on individuals' health and wellbeing. By prioritizing the health and safety of staff, organizations like UNRWA can ensure that they are better equipped to respond to future health crises and to support their employees' overall health and wellbeing.

Globally, the precise definition, risk assessment tools, etiology, and risk factors of Post COVID-19 syndrome are still debatable, and we are in need to establish a reference.

## Supporting information

**S1 Data.**
(XLSX)

## Acknowledgments

Prof. Anwar Batieha passed away before the submission of the final version of this manuscript. Dr. Haneen Aldahleh accepts responsibility for the integrity and validity of the data collected and analyzed. His contributions to research and education have left an indelible mark. May his legacy continue to inspire future generations in the field. We are grateful for all UNRWA staff who participated in this study.

## Author Contributions

**Conceptualization:** Haneen Aldahleh, Anwar Batieha.

**Data curation:** Haneen Aldahleh.

**Formal analysis:** Haneen Aldahleh.

**Methodology:** Haneen Aldahleh.

**Supervision:** Anwar Batieha.

**Validation:** Rasheed Elayyan.

**Writing – original draft:** Haneen Aldahleh.

**Writing – review & editing:** Rasheed Elayyan, Nour Abdo, Ishtaiwi Abuzayed, Shatha Albaik, Yousef Shahin, Akihiro Seita.

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
