## [Decision Letter · Decision Letter 0]

27 Mar 2023

PONE-D-23-03062Clinical Profile, Prognosis and Post COVID-19 Syndrome among UNRWA Staff in Jordan: A Clinical Case-Series Study

PLOS ONE

Dear Dr. Aldahleh,

Thank you for submitting your manuscript to PLOS ONE. After careful consideration, we feel that it has merit but does not fully meet PLOS ONE’s publication criteria as it currently stands. Therefore, we invite you to submit a revised version of the manuscript that addresses the points raised during the review process.

ACADEMIC EDITOR COMMENTS:

Dear authors,

Thank you for your submission to PLoS One. 

I have reviewed your manuscript, along with the external peer review we have been able to solicit thus far. 

On this basis, and while we await further peer reviews, I want to offer an initial response. The reason for this is that my assessment, and that of the first peer reviewer, suggests some major changes / elaboration is required in order for your manuscript to be considered for publication. Rather than delaying further, I want to offer this initial decision to move this along.

My comments are as follows:

Line 58 - "was reported in Wuhan"

Line 66 - "differs" 

Lines 68 - 71 - several studies have now explored Long COVID. Are there any from Jordan, or closer in the region, that would allow for more country- / region-specific comparison? You mention no such studies, but I had a quick look and saw this pre-print paper: https://www.emro.who.int/in-press/research/post-covid-syndrome-among-jordanian-health-care-workers.html, and this: xhttps://www.iproc.org/2022/1/e36563/

Line 77 - 78 - is this also a study of risk factors? 

Line 93 - as Reviewer 1 has mentioned, this requires elaboration. The sample frame is clear - i.e. all known positive staff (presumably lab tested? - perhaps you can also elaborate how a positive diagnosis was determined). But the sampling strategy is not. How were these people selected? 

Line 93 - 94 - might be worth explaining why those with a recent diagnosis were excluded - perhaps a definition of long COVID earlier in the manuscript. 

Line 107 - can you elaborate what the pilot study involved? How was validation achieved? 

Line 233 - 237 - some negative findings might be worth presenting. Did smoking status have no other statistically significant impact on other symptoms? Severity of illness, SOB, etc? 

Line 265 - "older people over the age of 60"

Line 271 - how were asymptomatic people identified? Did UNRWA have a testing system in place at that time?

Line 285 - I don't think there's any strong basis with which to suggest symptom profiles differed by race and ethnicity

Line 319 - 320 - here you mention a Jordan-based long COVID study, which runs counter to your earlier suggestion of no pre-existing country research

Line 372 - people of working age (broad as that is 18-65ish) can have high rates of HTN, T2DM - how does this compare to the general Jordanian population?

Line 388 - again, how this is a representative sample needs to be elaborated, as the sampling strategy is unclear

Line 395 - a major limitation here is that this study didn't have a longitudinal dimension. So symptoms reported have been attributed to COVID, but we don't know if people had pre-existing fatigue, forgetfulness, depression, etc. 

Line 402 - Conclusion should not reiterate results, but take your Discussion and wrap-up with next steps, what this adds, why it makes any difference, who should take note, etc.

We look forward to receiving your revised manuscript.

Kind regards,

James Smith

Academic Editor

PLOS ONE

Journal Requirements:

"No"

"No"

Reviewers' comments:

Reviewer's Responses to Questions

**Comments to the Author**

1. Is the manuscript technically sound, and do the data support the conclusions?

Reviewer #1: Partly

2. Has the statistical analysis been performed appropriately and rigorously? 

Reviewer #1: I Don't Know

3. Have the authors made all data underlying the findings in their manuscript fully available?

Reviewer #1: No

4. Is the manuscript presented in an intelligible fashion and written in standard English?

Reviewer #1: Yes

5. Review Comments to the Author

Reviewer #1: Thank you for the opportunity to review this interesting paper. There are some major concerns I think important to clarify and address. Most importantly, it is unclear how study participants were selected. The authors state "We selected our sample (366 out of 1322 cases) systemically." What do they mean exactly? On what systematic basis did they select participants, this needs to be clearly outlined in the paper. Further, it is unclear if a simple chi-square test is sufficient to fully describe the association between sociodemographic and clinical covariates and outcomes. For example, could it be that the sample that received 2 doses were different in terms of their clinical background risk than those who only took 1 vaccination? A multivariate model would have helped explain some of these differences. The great value of this paper is in the descriptives, but to fully appreciate it is important to clarify the above.

6. PLOS authors have the option to publish the peer review history of their article (what does this mean?). If published, this will include your full peer review and any attached files.

Reviewer #1: No

---

## [Author Response · Author response to Decision Letter 0]

16 May 2023

Re: Clinical Profile, Prognosis and Post COVID-19 Syndrome among UNRWA Staff in Jordan: A Clinical Case-Series Study

Dear Reviewers,

I hope this letter finds you well. First and foremost, I would like to express my gratitude for taking the time to review my publication. Your constructive feedback and comments have been invaluable in improving the quality and clarity of my work.

I would like to address the concerns and comments raised in your review. I have carefully considered each one of them and have made the necessary changes to the manuscript accordingly.

So please find the following comments and responses. 

Line 58 - "was reported in Wuhan"

Changed according to reviewer comment.

Line 66 - "differs"

Changed according to reviewer comment.

Lines 68 - 71 - several studies have now explored Long COVID. Are there any from Jordan, or closer in the region, that would allow for more country- / region-specific comparison? You mention no such studies, but I had a quick look and saw this pre-print paper: https://www.emro.who.int/in-press/research/post-covid-syndrome-among-jordanian-health-care-workers.html, and this: xhttps://www.iproc.org/2022/1/e36563/: 

This was reviewed now and justified on the manuscript, in addition to this pre-print we found another study also. 

Line 77 - 78 - is this also a study of risk factors? 

No it is not, we made the correction on the manuscript.

Line 93 - as Reviewer 1 has mentioned, this requires elaboration. The sample frame is clear - i.e. all known positive staff (presumably lab tested? - perhaps you can also elaborate how a positive diagnosis was determined). But the sampling strategy is not. How were these people selected? 

Sample strategy is now clarified in more details on the manuscript. 

Line 93 - 94 - might be worth explaining why those with a recent diagnosis were excluded - perhaps a definition of long COVID earlier in the manuscript. 

Exclusion criteria according to post COVID definition is now clarified on the manuscript. 

Line 107 - can you elaborate what the pilot study involved? How was validation achieved? 

We explained more about the pilot study now on the manuscript.

Line 233 - 237 - some negative findings might be worth presenting. Did smoking status have no other statistically significant impact on other symptoms? Severity of illness, SOB, etc? 

Some related negative findings added. 

Line 265 - "older people over the age of 60"

Changed according to reviewer comment.

Line 271 - how were asymptomatic people identified? Did UNRWA have a testing system in place at that time?

According to our country policy at that time, any person who had a contact with infected person should do the PCR test, so some employees contacted infected people while they are free of symptoms. Staff who were contacting their infected symptomatic relatives, were reporting this to their work in which they will be asked to conduct quarantine and testing accordingly. So they inform UNRWA and being part of its record of infected employees

Line 285 - I don't think there's any strong basis with which to suggest symptom profiles differed by race and ethnicity

This is removed from the Manuscript-Not Justified and we agree with the reviewer.

Line 319 - 320 - here you mention a Jordan-based long COVID study, which runs counter to your earlier suggestion of no pre-existing country research

We added the pre-print and the study that we found now earlier in the manuscript. 

Line 372 - people of working age (broad as that is 18-65ish) can have high rates of HTN, T2DM - how does this compare to the general Jordanian population?

We compare this now with the general population figures.

Line 388 - again, how this is a representative sample needs to be elaborated, as the sampling strategy is unclear

We elaborate on sample selection in the methods section earlier now. 

Line 395 - a major limitation here is that this study didn't have a longitudinal dimension. So symptoms reported have been attributed to COVID, but we don't know if people had pre-existing fatigue, forgetfulness, depression, etc. 

According to the definition of post COVID19 syndrome WHO, Symptoms were considered positive only if persisted for 2 months or more after 12 weeks from date of infection and the symptom cannot be explained by an alternative diagnosis. we focused when we conducted the questionnaire that the symptom was newly developed and no other diagnosis explained it, otherwise it would be considered negative.

Line 402 - Conclusion should not reiterate results, but take your Discussion and wrap-up with next steps, what this adds, why it makes any difference, who should take note, etc.

The conclusion section is now modified according to your advice.

Once again, I would like to thank you for your time and effort in reviewing my publication. Your insights and suggestions have been invaluable, and I believe that they have greatly improved the quality of my work. If you have any further comments or suggestions, please do not hesitate to let me know.

Sincerely,

Haneen Aldahleh

---

## [Decision Letter · Decision Letter 1]

4 Sep 2023

PONE-D-23-03062R1Clinical Profile, Prognosis and Post COVID-19 Syndrome among UNRWA Staff in Jordan: A Clinical Case-Series StudyPLOS ONE

Dear Dr. Haneen Yousef Aldahleh,

Thank you for submitting your manuscript to PLOS ONE. After careful consideration, we feel that it has merit but does not fully meet PLOS ONE’s publication criteria as it currently stands. Therefore, we invite you to submit a revised version of the manuscript that addresses the points raised during the review process.

We look forward to receiving your revised manuscript.

Kind regards,

Fadi Aljamaan

Academic Editor

PLOS ONE

Journal Requirements:

Additional Editor Comments (if provided):

Dear Dr Haneen Yousef Aldahleh,

Please revise your manuscript and respond to the most recent comments that were given based on your last draft.

Reviewers' comments:

Reviewer's Responses to Questions

**Comments to the Author**

1. If the authors have adequately addressed your comments raised in a previous round of review and you feel that this manuscript is now acceptable for publication, you may indicate that here to bypass the “Comments to the Author” section, enter your conflict of interest statement in the “Confidential to Editor” section, and submit your "Accept" recommendation.

Reviewer #1: All comments have been addressed

Reviewer #2: All comments have been addressed

2. Is the manuscript technically sound, and do the data support the conclusions?

Reviewer #1: Yes

Reviewer #2: Partly

3. Has the statistical analysis been performed appropriately and rigorously? 

Reviewer #1: I Don't Know

Reviewer #2: Yes

4. Have the authors made all data underlying the findings in their manuscript fully available?

Reviewer #1: No

Reviewer #2: Yes

5. Is the manuscript presented in an intelligible fashion and written in standard English?

Reviewer #1: Yes

Reviewer #2: Yes

6. Review Comments to the Author

Reviewer #1: Thank you for the opportunity to review the revised manuscript. You have addressed all the comments.There have been insufficient studies of the impact of COVID-19 on long term health outcomes and this is a meaningful contribution to address this gap in the research.

Reviewer #2: Dear Authors:

I have few comments that need to be addressed in order to validate your results.

1. you mentioned that 1322 of the staff developed COVID-19 while your sample size is only 366, I believe these are your colleagues and easily reachable, therefore why did you limit the sample to quarter of the eligible participants.

2. As you mentioned and is agreed on, the post COVID syndrome Sx should be lasting for at least 2 months post acute infection but WITH NO ALTERNATIVE DIAGNOSIS FOUND, how did you make sure of that?

2. Regarding the post COVID Sx that were found associated with obesity, many of them are actually related to OSA symptoms, do you have any clarification?

3. Your discussion of the results is superficial and doesn't give possible explanations for the association you found, you only mention evidence from literature about your results.

4. You don't need to repeat your results in details especially numbers in the discussion section.

7. PLOS authors have the option to publish the peer review history of their article (what does this mean?). If published, this will include your full peer review and any attached files.

Reviewer #1: No

Reviewer #2: **Yes: **fadi aljamaan

---

## [Author Response · Author response to Decision Letter 1]

3 Oct 2023

1. you mentioned that 1322 of the staff developed COVID-19 while your sample size is only 366, I believe these are your colleagues and easily reachable, therefore why did you limit the sample to quarter of the eligible participants.

The decision to limit our sample size was based on both scientific considerations and practical logistics. it might seem straightforward to engage with colleagues for data, the constraints of resources, combined with the detailed nature of data collection – which included not just the time of the researcher but also the expenses associated with hiring additional data collectors to support our study – made it more practical to work with a smaller, concentrated sample. The logistics of interacting with a vast number of participants, especially via phone-based surveys conducted during working hours, further justified our approach. This method not only streamlined our data quality checks but also expedited the results, a crucial factor given the ongoing pandemic. Our choice of the 366 participants was methodically made using randomized sampling, ensuring they accurately represented the broader affected group's characteristics. In essence, our goal was to harmonize the acquisition of dependable and prompt data with the efficient use of our resources.

2. As you mentioned and is agreed on, the post COVID syndrome Sx should be lasting for at least 2 months post acute infection but WITH NO ALTERNATIVE DIAGNOSIS FOUND, how did you make sure of that?

Since our staff are insured, the COVID-19 team was actively monitoring them throughout their infection and subsequent recovery. We consistently urged them to report any lingering symptoms and to consult specialists if these symptoms continued to impact their quality of life. Concurrently, the medical history gathered during our study was comprehensive enough to confirm that the enduring symptoms they reported were indeed new and consistent with post-COVID-19 effects.

3. Regarding the post COVID Sx that were found associated with obesity, many of them are actually related to OSA symptoms, do you have any clarification?

While obesity is a known risk factor for both post-COVID symptoms and obstructive sleep apnea , it's vital to distinguish between the two given their symptom overlap. In our study, we differentiated symptoms unique to post-COVID syndrome, such as anosmia or ageusia, from those commonly seen in OSA. By examining participants' prior medical histories and, we could discern the origin of certain symptoms. We also assessed the timing of symptom onset, considering whether issues like fatigue predated a COVID-19 infection.

4. Your discussion of the results is superficial and doesn't give possible explanations for the association you found; you only mention evidence from literature about your results.

You don't need to repeat your results in details especially numbers in the discussion section.

I update this section by adding the followings:

• The symptomatic profile we observed, dominated by fatigue, smell impairment, and fever, resonates with international findings but with certain nuances. Such variations could stem from differences in genetic predispositions, viral strains, or environmental factors, warranting further investigation.

• While it's recognized that post-COVID symptoms might be influenced by factors like age, gender, or comorbidities, our study interestingly identified associations with smoking. This aligns with some international studies but diverges from others, hinting at potential genetic, behavioral, or regional factors at play.

Many thanks for your valuable comments.

We are looking forward to hearing from you.

---

## [Editor Report · Decision Letter 2]

4 Oct 2023

Clinical Profile, Prognosis and Post COVID-19 Syndrome among UNRWA Staff in Jordan: A Clinical Case-Series Study

PONE-D-23-03062R2

Dear Dr. Haneen Yousef Aldahleh,

We’re pleased to inform you that your manuscript has been judged scientifically suitable for publication and will be formally accepted for publication once it meets all outstanding technical requirements.

Kind regards,

Fadi Aljamaan

Academic Editor

PLOS ONE
---

## [Editor Report · Acceptance letter]

23 Oct 2023

PONE-D-23-03062R2 

Clinical Profile, Prognosis and Post COVID-19 Syndrome among UNRWA Staff in Jordan: A Clinical Case-Series Study 

Dear Dr. Aldahleh:

I'm pleased to inform you that your manuscript has been deemed suitable for publication in PLOS ONE. Congratulations! Your manuscript is now with our production department. 

Kind regards, 

on behalf of

Dr. Fadi Aljamaan 

Academic Editor

PLOS ONE